

# Validation of system usability scale as a usability metric to evaluate voice user interfaces

Akshay Madhav Deshmukh and Ricardo Chalmeta

Systems Integration and Re-Engineering Group (IRIS), Department of Computer Languages and Systems, Universitat Jaume I de Castellón, Castello de la Plana, Castellon, Spain

## ABSTRACT

In recent years, user experience (UX) has gained importance in the field of interactive systems. To ensure its success, interactive systems must be evaluated. As most of the standardized evaluation tools are dedicated to graphical user interfaces (GUIs), the evaluation of voice-based interactive systems or voice user interfaces is still in its infancy. With the help of a well-established evaluation scale, the System Usability Scale (SUS), two prominent, widely accepted voice assistants were evaluated. The evaluation, with SUS, was conducted with 16 participants who performed a set of tasks on Amazon Alexa Echo Dot and Google Nest Mini. We compared the SUS score of Amazon Alexa Echo Dot and Google Nest Mini. Furthermore, we derived the confidence interval for both voice assistants. To enhance understanding for usability practitioners, we analyzed the Adjective Rating Score of both interfaces to comprehend the experience of an interface's usability through words rather than numbers. Additionally, we validated the correlation between the SUS score and the Adjective Rating Score. Finally, a paired sample t-test was conducted to compare the SUS score of Amazon Alexa Echo Dot and Google Nest Mini. This resulted in a huge difference in scores. Hence, in this study, we corroborate the utility of the SUS in voice user interfaces and conclude by encouraging researchers to use SUS as a usability metric to evaluate voice user interfaces.

## INTRODUCTION

The use of voice assistants has been rising in recent years. Over 52% of users prefer voice assistants over websites for information searches (*Laricchia, 2022a*). Current research states that the use of voice assistants depends greatly on their hedonic capabilities and degree of usability (*Pal et al., 2020*; *McLean & Osei-Frimpong, 2019*). As of 2019, the global voice-assistant market was valued at USD 11.9 billion and is anticipated to increase to USD 35.5 billion by the year 2025 (*Laricchia, 2022b*). Among the many popular voice assistants, Amazon Alexa Echo Dot and Google Nest Mini were examined in this study.

The choice of Amazon Alexa Echo Dot and Google Nest Mini was backed by a strong market research study showing that Amazon is the most popular, leading vendor in the global smart speaker market, with a market share of 26.4% in Q3 of 2021. This translates to

Corresponding author
Ricardo Chalmeta, rchalmet@uji.es

10.3 million devices sold in the same quarter. Amazon's closest competitor is Google, with a market share of 20.5% in Q3 of 2021. This represents 8.1 million devices sold in the same quarter (*Laricchia, 2023*; *Michail, 2021*).

As a pioneer of usability, Jakob Nielsen noted: "*Usability rules the web. Simply stated, if the customer can't find a product, then he or she will not buy it*" (*Nielsen, 1999*). Usability means making technology that works for people. It "is a quality attribute that assesses how easy user interfaces are to use." Alternatively, it refers to the "quality of the interaction in terms of parameters such as time taken to perform tasks, the number of errors made, and the time to become a competent user" (*Benyon, 2014*; *Feng & Wei, 2019*). From a user's perspective, usability plays a huge role in the development process as it makes a difference in the performance and completion of a task successfully, without any exasperation.

In the current research, we can find a lot of literature on usability measurements for GUIs. However, there is a huge gap in research on the usability measurements for voice assistants and speech-based interfaces. Currently, there are no standard or well-established metrics for measuring the usability of voice assistants or voice-enabled devices (*Kocabalil, Laranjo & Coiera, 2018*; *Lewis, 2016*). Due to the lack of a standardized usability measure, it is difficult to assess voice-based devices meaningfully. This study is an attempt to narrow the gap in the existing literature by considering a well-established tool, the System Usability Scale (SUS), to assess the usability of Amazon Alexa Echo Dot and Google Nest Mini. The main goal is to determine whether SUS can be used as a usability metric or a tool to evaluate voice-based user interfaces.

Several questionnaires have been developed to evaluate the user's understanding of an interface. Questionnaires have been used widely to evaluate user interfaces in a system that *Tullis & Albert (2013)* designed to evaluate different aspects of usability. However, a wide range of problems have been encountered. Most of the subjective evaluation measurement tools were found to be very feeble (*Chin, Norman & Shneiderman, 1987*; *Ives, Olson & Baroudi, 1983*). There were problems ranging from insufficiency of validation (*Gallagher, 1974*) to unreliability (*Larcker & Lessig, 1980*). There were also problems with participants choosing the same options for most of the questions which made reliability a huge concern (*Ives, Olson & Baroudi, 1983*). A vast amount of research has been undertaken to examine the type of questions that would be suitable for questionnaires. Questionnaires in the form of checklists were not effective in the evaluation of systems, as the need for new features was not evident (*Root & Draper, 1983*). Hence, the questionnaires in the form of checklists have been supplemented with open-ended types of questionnaires (*Coleman, Williges & Wixon, 1985*).

There are many usability evaluation tools available in addition to the SUS (*Brook, 1996*). Examples are the Questionnaire for User Interface Satisfaction (QUIS; *Chin, Diehl & Norman, 1988*), Perceived Usefulness and Ease of Use (PUEU; *Davis, 1989*), Nielsen's Attributes of Usability (NAU; *Nielsen, 1994a*), Nielsen's Heuristic Evaluation (NHE; *Nielsen, 1994b*), Computer System Usability Questionnaire (CSUQ; *Lewis, 1995*), After Scenario Questionnaire (ASQ; *McIver & Carmines, 1981*; *Nunnally & Bernstein, 1978*),

| # | Item |
|---|---|
| | **Table 1 System usability scale.** |
| 1 | I think that I would like to use this interface frequently. |
| 2 | I found the interface unnecessarily complex. |
| 3 | I thought the interface was easy to use. |
| 4 | I think that I would need the support of a technical person to be able to use this interface. |
| 5 | I found the various functions in the interface were well integrated. |
| 6 | I thought there was too much inconsistency in this interface. |
| 7 | I imagine that most people would learn to use this interface very quickly. |
| 8 | I found the interface very cumbersome to use. |
| 9 | I felt very confident using the interface. |
| 10 | I needed to learn a lot of things before I could get going with this interface. |

Practical Heuristics for Usability Evaluation (PHUE; *Perlman & Gary, 1995*), and the Software Usability Measurement Inventory (SUMI; *Kulkarni et al., 2013*).

In this study, we chose SUS for the following reasons:

1. According to market research (*Lewis, 2018*), SUS has been used across many industries in over 43% of their usability studies. It is one of the most popular usability tools that has been used recently. According to a study in *Peres, Pham & Phillips (2013)*, SUS has been cited 9,516 times, which is an appropriate metaphor for its popularity.

2. According to the research, SUS is more reliable, valid and sensitive to a range of independent variables (*Bangor, Kortum & Miller, 2008*).

3. The SUS scale is considered a more universal scale that is independent of the technology. This provides the opportunity to explore applicability in voice-enabled devices (*Lewis, 2018*; *Bangor, Kortum & Miller, 2008*).

For the above reasons, SUS is a good candidate for usability evaluation. However, there are also some concerns. According to *Bangor, Kortum & Miller (2009)*, SUS is primarily used to evaluate GUI-based systems, and some items in the SUS may not be justified for VUI's. The interface of the latest voice assistant platforms like Amazon Alexa Echo Dot and Google Nest Mini has no visual feedback. This increases the users' cognitive load (*Ghosh et al., 2018*; *Luger & Sellen, 2016*). In turn, the absence of visual feedback leads to poor learnability of what a system does among users.

Table 1 shows a SUS questionnaire where, only for the purpose of this study, the word "interface" has been used interchangeably with the word "system" (from the original SUS questionnaire) for better interpretation. In the SUS questionnaire, items 2, 6 and 10 tend to be more relevant for GUI as they possess more visual feedback than voice-based interfaces. This raises concern about applicability to VUIs. Hence, this study aimed to determine whether SUS can be used as a usability metric or tool to evaluate voice-based interfaces.

**Table 2 Set of tasks to be performed by participants on Amazon Alexa Echo Dot and Google Nest Mini.**

| # | Tasks |
|---|---|
| 1 | How is the weather today in Berlin |
| 2 | What is the latest news for today |
| 3 | Play workout music |
| 4 | Play Radiohead playlist. |
| 5 | Jump to the next song in the playlist. |
| 6 | Increase the volume. |
| 7 | Add an item to the grocery list. |
| 8 | Set alarm for tomorrow at 5 a.m. |
| 9 | Search Wikipedia for Elon Musk. |
| 10 | What is Coronavirus. |

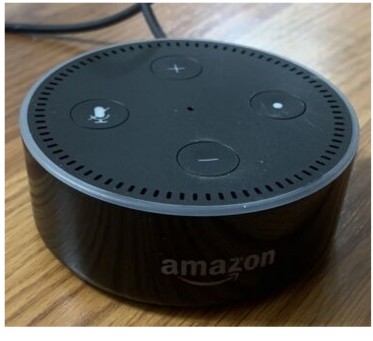    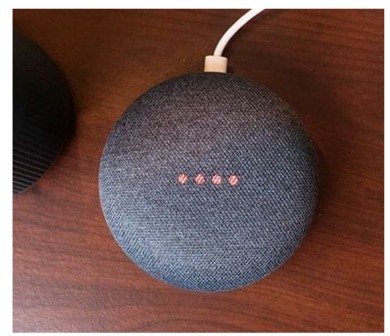

(a) Amazon Echo Dot          (b) Google Nest Mini

**Figure 1 Device under test (DUT).** Image credit: Akshay Deshmukn.

# METHODOLOGY: USABILITY EVALUATION

## Participant recruitment

In the study, we recruited 16 participants (seven female and nine male) from universities and research organizations. Their *mean age was = 28.43 years (SD = 3.48)*. The inclusion criteria for participant recruitment were:

1. They do not own an Alexa or similar devices at home.
2. They do not have any prior experience of using any voice-based smart assistants, especially Amazon Alexa Echo Dot and Google Nest Mini.
3. The participants have fluent English-speaking skills to proficiency level.

## Procedure

The participants were invited and given a set of ten tasks to be performed. Table 2 describes these tasks. The study was conducted in person with moderation.

The device under test (DUT's) were Amazon Alexa Echo Dot (Fig. 1A) and Google Nest Mini (Fig. 1B).

11. Overall, I would rate the user-friendliness of this voice interface as:

| ☐ | ☐ | ☐ | ☐ | ☐ | ☐ | ☐ |
|---|---|---|---|---|---|---|
| Worst Imaginable | Awful | Poor | OK | Good | Excellent | Best Imaginable |

**Figure 2  Adjective rating scale.**

To avoid preferential biases of one device over the other, we followed best practices according to *Dumas & Fox (2007)*. Out of 16 participants, eight participants started the usability study with Amazon Alexa Echo Dot and the other eight participants started their usability study with Google Nest Mini.

After completion of the set of tasks for each interface, participants were given a form with an SUS questionnaire of 10 items. Each item in the SUS questionnaire was rated from 1 to 5 on the Likert scale (1 = strongly disagree and 5 = strongly agree).

To calculate the SUS score, we calculated the sum of the score contributions for each item. Each item's score ranged from 0 to 4. For items 1, 3, 5, 7 and 9 (odd-numbered items) the score contribution was the scale position minus 1. For items 2, 4, 6, 8 and 10 (even-numbered items) the score contribution was 5 minus the scale position. Next, we multiplied the sum of scores by 2.5 to obtain the overall System Usability value. SUS scoresrange from 0 to 100 (*Brook, 1996*).

Along with the 10-item SUS questionnaire, there was an additional Adjective Rating Scale statement at the end of the form. Participants were advised to fill it in immediately after SUS ratings were made. The Adjective Rating Scale answers the most obvious question asked by usability practitioners "What is the absolute usability associated with any SUS individual score?

The Adjective Rating Scale is a seven-point scale that couples SUS in a more pragmatic way. Numbers 1 through 7 were assigned to adjectives ranging from "worst imaginable" to "best imaginable" respectively (Fig. 2). These words or phrases help practitioners to gain better understanding of the usability aspect of SUS scores (*Bangor, Kortum & Miller, 2009*).

## RESULTS

The results can be organized in three categories: SUS score and confidence interval (CI), Adjective Rating Scale for both interfaces, and Adjective Rating Scale correlation with SUS score.

### SUS score and confidence interval

We begin the SUS score analysis by comparing the mean SUS scores of the Amazon Echo Dot and Google Nest Mini. With five questions with positive statements and five with negative ones, the SUS is a well-balanced, well-designed survey with scores ranging from 0 to 100.

The results shows that there is a considerable difference between the mean SUS score for the two assistants. Figure 3 shows that, for Alexa, the mean SUS score is 81.623 and the
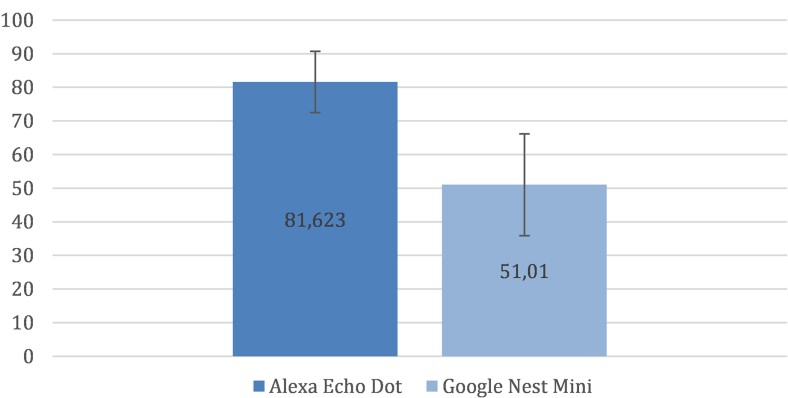

**Figure 3  Mean SUS Score with standard deviation.**

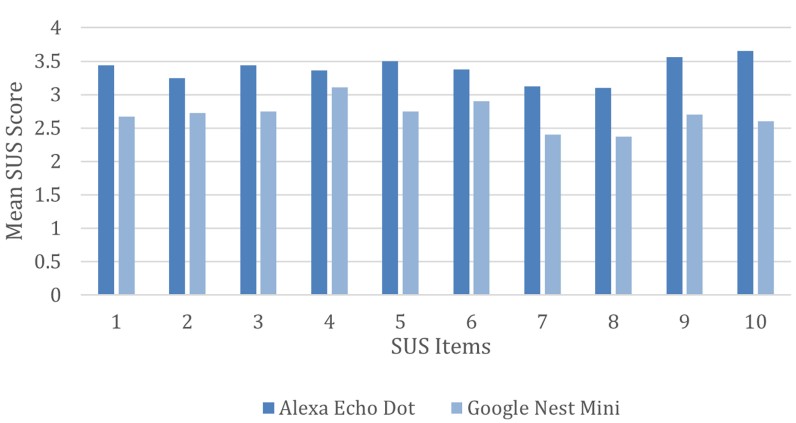

**Figure 4  Mean SUS score for all 10 items.**

standard deviation (SD) is 9.12 and for Google Nest Mini the mean SUS score is 51.01 and the SD is 15.11.

A study by *Sauro & Lewis (2016)*, who used data from 241 industrial usability studies and surveys to create a curved grading scale, found that the mean SUS score for interfaces was 68, with 50% of interfaces falling below and above it. Therefore, a mean SUS score > 80 can be considered semantically "good" and achieving it should be a common industrial goal as evidence of an above average user experience. A mean SUS score < 70 can be considered semantically as "usability issues with a cause of concern" (*Bangor, Kortum & Miller, 2008*; *Lewis & Sauro, 2018*). This implies that Alexa (with a mean SUS score > 80) performed above average and Google Nest Mini (with a mean SUS score < 70) performed lower than the set standards.

Furthermore, the mean SUS score was calculated for all the 10 individual items for both interfaces. As shown in Fig. 4, Alexa performed way better than Google Nest Mini on all the SUS items. Alexa had a mean SUS score of above 3.4 for items 1, 3, 5, 9 and 10 and had

a low score for items 7 and 8 with a mean SUS score just above three. On the other hand, Google Nest Mini obtained a mean SUS score for items 4 and 6 of just above three and the lowest SUS score for items 7 and 8 with under 2.5.

Furthermore, we calculated the CI for the available sample size (total number of participants) in the following way:

$$CI = \bar{X} \pm Z \times \frac{\sigma}{\sqrt{n}}$$

where $n$ is the sample size or the total number of participants involved in the study = 16, $\bar{X}$ is the mean SUS score of Alexa and Google Nest Mini, which is 81.623 and 51.01 respectively, and $\sigma$ is the Standard Deviation of Alexa and Google Nest Mini, which is 9.12 and 15.11 respectively. Considering the confidence level to be 95%, its corresponding Z value was interpreted as 1.960 according to *Moore (1996)*.

Therefore, the CI for Amazon Alexa is 77.154–86.092 with the confidence value of ±4.469 (error percentage ±5.5%) and the CI for Google Nest Mini is 43.606–58.414 with the confidence value of ±7.404 (error percentage ±14.5%). Figure 5 shows these results.

Hence, from Fig. 5, we can infer that the Alexa Echo Dot results have less uncertainty, with an error margin of ±5.5%, compared to Google Nest Mini results, which have an error margin of ±14.5%. Consequently, it seems that the Alexa Echo Dot results are more reliable that the Google Nest Mini results.

## Adjective rating scale

The Adjective Rating Scale is on the absolute usability associated with the SUS individual score. This scale provides comprehendible words or small phrases that can be associated with a range of SUS scores.

Figure 6 shows the Adjective Rating Scale of each participant for both interfaces. All the participants felt that the user friendliness of the Alexa Echo Dot was much better than Google Nest Mini, except for two participants who felt that the user friendliness of both interfaces was the same. The majority of participants (numbers 1, 2, 3, 4, 5, 6 and 10) had a significant difference in the adjective rating, which further implies that the user friendliness of Alexa stood out compared to Google Nest Mini.

On the other hand, Table 3 shows the total number of counts for each Adjective Rating Scale from all the participants for Alexa Echo Dot and Google Nest Mini. Notably, all the participants rated Alexa with positive adjectives while Google received negative, neutral and positive adjectives.

## Adjective rating scale correlation

After the SUS and Adjective Rating Scale analysis for Alexa Echo Dot and Google Nest Mini, we conducted an analysis of the correlation between both interfaces (convergent validity between SUS and Adjective Rating Scale). The finding is that the SUS scores and Adjective Rating Scales were strongly correlated with each other with a Pearson's correlation coefficient of r = 0.9093 (Fig. 7).

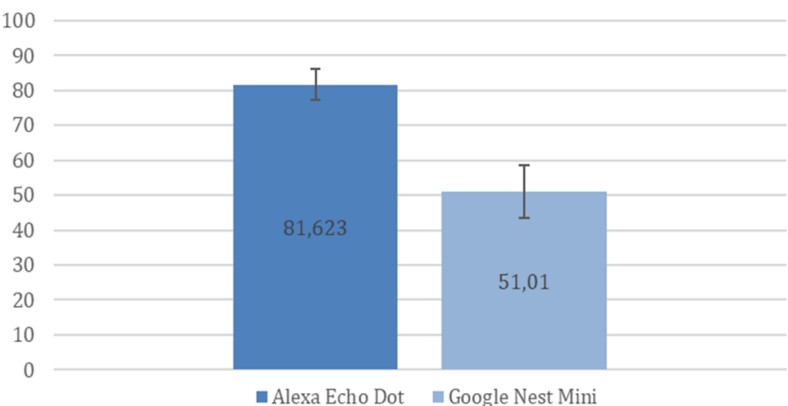

**Figure 5** Mean SUS score with confidence interval.

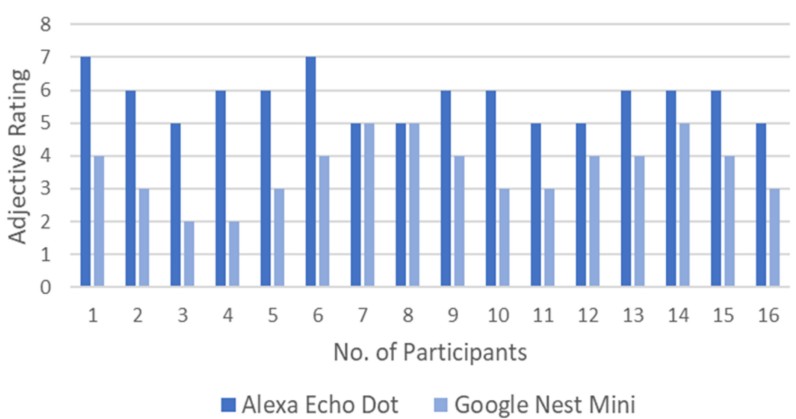

**Figure 6** Adjective Rating Score by all participants for both interfaces.

**Table 3 Number of counts of each adjective rating scale.**

| Adjective rating scale | No. of counts (Alexa Echo Dot) | No. of counts (Google Nest Mini) |
|---|---|---|
| 1 (Worst imaginable) | 0 | 0 |
| 2 (Awful) | 0 | 2 |
| 3 (Poor) | 0 | 5 |
| 4 (OK) | 0 | 6 |
| 5 (Good) | 6 | 3 |
| 6 (Excellent) | 8 | 0 |
| 7 (Best imaginable) | 2 | 0 |

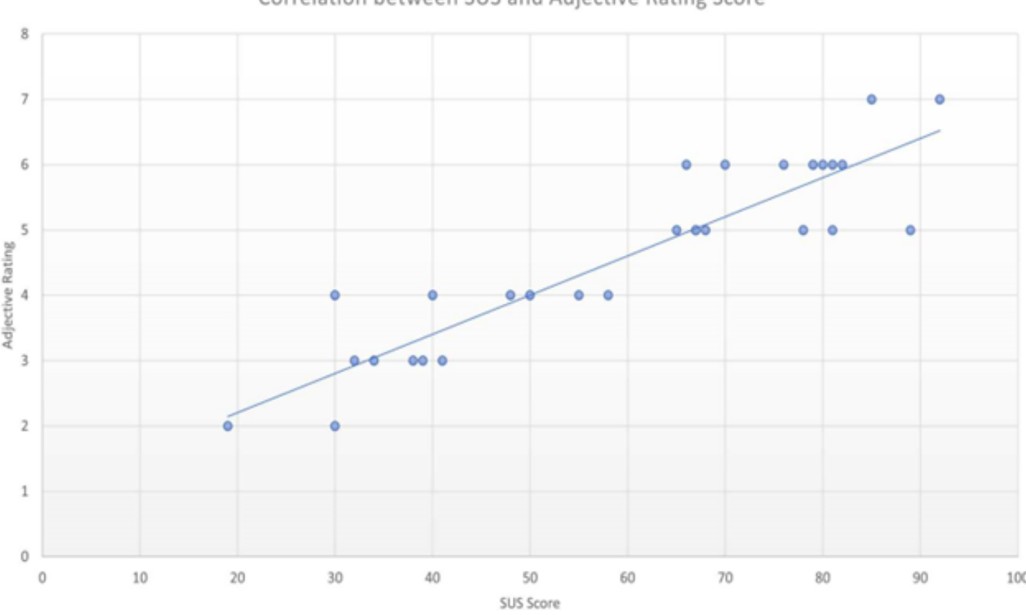

**Figure 7 Correlation between SUS and adjective rating scale (Pearson's correlation coefficient r = 0.9093).**

Finally, with the total sample, which in this case included 16 participants with a mean SUS score of = 81.623 and SD of = 9.12 for Amazon Alexa Echo Dot and a mean SUS score of = 51.01 and SD of = 15.11 for Google Nest Mini, we conducted a paired sample t-test.

$$t = \frac{(\bar{X}_1 - \bar{X}_2)}{\sqrt{\left(\frac{S_1^2}{n_1} + \frac{S_2^2}{n_1}\right)}}$$

where, $\bar{X}_1$ and $\bar{X}_2$ are the mean SUS score of Amazon Alexa Echo Dot and Google Nest Mini respectively; $S_1^2$ and $S_2^2$ are the variances of Alexa Echo Dot and Google Nest Mini respectively; and $n_1$ and $n_2$ are the total sample size (number of participants).

The result of the t-statistic was t = 0.92 and the probability value (*p*) associated with the t-statistic is *p* = 0.00001.

The means of two measurements made from the same subject, item, or related units were compared using the Paired Samples t test. The test's goal was to ascertain whether there is statistical support for the claim that the mean difference between paired observations differs noticeably from zero. The Paired Samples t test is a parametric test.

# DISCUSSION

## Contributions to the state of the art of human-computer interfaces

First, we applied the SUS score to two voice user interfaces: Amazon Alexa Echo Dot and Google Nest Mini. Although SUS has been designed to be applied to all types of systems

(*Gronier & Baudet, 2021*) such as augmented reality (*Hatzl et al., 2023*), websites (*Igarashi, Kobayashi & Nihei, 2023*), mobile applications (*Moorthy et al., 2023*), expert systems (*Azmi et al., 2023*), serious games (*Höhler et al., 2023*) and e-learning systems (*Heinzel et al., 2020*), in the literature there is a lack of studies that apply SUS to voice users interfaces. Therefore, we validated the System Usability Scale as a metric to evaluate voice users interfaces, an emerging field.

Second, we analyzed the Adjective Rating Score of both interfaces and validated the correlation between the SUS score and the Adjective Rating Scale. The Pearson's correlation coefficient of our study is 0.9093 which is on a par with the score of 0.86 found in a study by *Bangor, Kortum & Miller (2008)*. Therefore, we proved the utility of the Adjective Rating Scale in helping to provide a subjective equivalence for an individual study's mean SUS score for voice user interfaces, a field where there is a scarcity of examples in the literature.

Third, we compared the SUS scores and the Adjective Rating Scale of two voice user interfaces. The Alexa Echo Dot had a better SUS score on all items compared to Google Nest Mini. Alexa turned out to be semantically "good" and Google semantically "average (OK)". With these results, we can infer that Alexa performed above average and Google Nest Mini performed lower than the user expectations. Therefore, we proved that SUS scores and the Adjective Rating Scale can be useful to compare the usability of different voice user interfaces.

## Implications for academics and practitioners

This study has several implications. On the one hand, academics could carry out more studies to prove that the correlation between the SUS score and the Adjective Rating Score for voice user interfaces usability evaluation shown in this work can be generalized. In addition, when SUS is applied to a voice user interface, it would be interesting to compare the results with established benchmarks, to contextualize the usability level relative to the standards. Academics could work on providing these standards. Finally, since the SUS relies on a scale response, which can be limiting in terms of granularity, participants may find it difficult to express nuanced opinions. This leads to skewed data points. Therefore, academics could work on analyzing the limitations of using such a scale and propose solutions and recommendations to overcome this challenge.

On the other hand, practitioners are encouraged to use SUS with the Adjective Rating Score as a usability metric to evaluate, identify strengths and weaknesses, compare and improve voice user interfaces.

## Limitations

Finally, it is important to state the study limitations. The main limitations to this study are potential biases and sample size. Although measures have been taken to avoid biases in the study design, six participants began with one interface and the other six participants began with the other interface, other biases could also have happened. Therefore, more research in this field is necessary to compare results.

## CONCLUSION

In this article, we compared the SUS score of Amazon Alexa Echo Dot and Google Nest Mini. We found that Alexa Echo Dot is semantically "good", with a better SUS score than Google Nest Mini. We also calculated the confidence interval of both interfaces. The error rate of Amazon Alexa was better than that of Google Nest Mini. In addition, we analyzed the Adjective Rating Score of both interfaces and validated the correlation between the SUS score and the Adjective Rating Score. The result of the correlation coefficient was on a par with the score of the previously conducted study. Finally, a paired sample t-test was carried out to compare the SUS score of Amazon Alexa Echo Dot and Google Nest Mini. A huge difference was found in their SUS score.

Hence, with this study, we can confirm the utility of the SUS in voice user interfaces and encourage researchers to use SUS as a usability metric to evaluate voice user interfaces.

### Funding

This work was supported by UJI 22I573.01/1 & GACUJIMC/2023/04. The funders had no role in study design, data collection and analysis, decision to publish, or preparation of the manuscript.

### Grant Disclosures

The following grant information was disclosed by the authors:
UJI 22I573.01/1 & GACUJIMC/2023/04.

### Competing Interests

The authors declare that they have no competing interests.

### Author Contributions

- Akshay Madhav Deshmukh conceived and designed the experiments, performed the experiments, analyzed the data, performed the computation work, prepared figures and/or tables, authored or reviewed drafts of the article, and approved the final draft.
- Ricardo Chalmeta conceived and designed the experiments, analyzed the data, prepared figures and/or tables, authored or reviewed drafts of the article, and approved the final draft.

### Data Availability

The raw data is available in the Supplemental File.

### Supplemental Information

Supplemental information for this article can be found online at http://dx.doi.org/10.7717/peerj-cs.1918#supplemental-information.

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
