# Peer review of "Validation of system usability scale as a usability metric to evaluate voice user interfaces"

_PeerJ Computer Science, doi:10.7717/peerj-cs.1918_

## Round 0.1 · original submission · Major Revisions

Based on the reviews received, I recommend the authors to make changes.

**Language Note:** PeerJ staff have identified that the English language needs to be improved. When you prepare your next revision, please either (i) have a colleague who is proficient in English and familiar with the subject matter review your manuscript, or (ii) contact a professional editing service to review your manuscript. PeerJ can provide language editing services - you can contact us at copyediting@peerj.com for pricing (be sure to provide your manuscript number and title). – PeerJ Staff

·

Basic reporting

The article deals with the applicability of the SUS (System Usability Scale) as a questionnaire for evaluating the usability of voice-based interface devices. Although the study proposal is interesting in its attempt to validate a well-recognized instrument for evaluating the usability of interfaces, there are some omissions in the text (listed below) that need to be clarified in order for the results to be valid.

Experimental design

Line 56 - there seems to me to be a typo in "de-developed"; also the sentence doesn't make much sense, because how can a questionnaire help the user to get to know a system? I think the sentence is confusing, because questionnaires are actually used to evaluate the user's understanding of the interface.

Line 92 - the paragraph says that the word system (original SUS) has been replaced by the word interface, but it's not clear in which situation. Was it for the study in question? I believe so, but it could have been made more explicit.

Line 100 - it's stated that one of the recruitment criteria was that the participants didn't own an Alexa-type device or similar, but I was left wondering what all the criteria used to recruit the participants were - all the recruitment criteria should be stated in the article.

Line 108 - how was the study designed (between or within subject)? I understood that it was within subject but it is not clear if the interaction with the devices was counterbalanced. Did all participants start the tasks with the same device? If so, this may have biased the study, as the participant is influenced by the second device in relation to their interaction with the first device. This study design needs to be better presented in the article.
It is difficult to trust the results presented without knowing the study design in detail.

Also, I was unsure whether the study was conducted online and self-administered or in person with moderation. Line 118 mentions an online form, which leads us to believe that the study may have been conducted remotely, but it's not clear whether it was moderated or not.

Line 117 - it was mentioned that "an additional "Adjective Rating Scale" statement added at the end of the online form", but no justification was given for the addition of this statement. What was the reason for this measurement?

Line 130 - what is the point and importance of presenting figure 3? These values presented in isolation have no meaning whatsoever, as the author himself (Brook, 1996) states in his publication. Nor is there any discussion of these figures in the article.

Validity of the findings

Due to the lack of information presented in the method and procedures used, it is difficult to analyze the comparison of results and the correlation between the scales. I suggest a major revision of this part of the article so that it can be better understood and prove the validity of the results.

Additional comments

No comments.

·

Basic reporting

The paper highlights the increasing significance of User Experience (UX) evaluation in interactive systems, specifically focusing on voice-based interfaces. It outlines the use of the System Usability Scale(SUS) to assess Amazon Alexa Echo Dot and Google Nest Mini, suggesting its adoption by researchers for such assessments. This review aims to provide constructive feedback on the manuscript.

All the figures have a strong relationship to the paper and experiment design. Each figure is well-labeled & described.

Figure 3: would be good to compare the mean SUS score with established benchmarks in the voice interface(if there are any) so readers could contextualize the usability level relative to the standards.
Would be good if the paper could also include the confidence interval as an indicator of the uncertainty associated with the mean SUS score
Raw data is supplied.

The authors conducted a detailed literature study. References are all complete.

Experimental design

The experiment's design is quite straightforward. The objective is concise and clear.

1. From lines 100 - 101: "...Some of the evaluation criteria for the participant recruitment were that they do not own an Alexa or similar devices at home." What is the full list of requirements when recruiting experiment participants? Are there any measures in place to ensure no potential biases were introduced?

2. Since the SUS relies on a scale response, which can be limiting in terms of granularity. Experiment participants may find it difficult to express nuanced opinions, leading to skewed data points. Is there anything future researchers should be aware of when evaluating a system using SUS? What could be the limitation of using such a scale?

Validity of the findings

The findings presented in this manuscript exhibit a high level of validity. While the simplicity in experimental design and tasks may contribute to ease of execution, the experiment design and results as well as the case study and t-test showed convincing evidence of how SUS could be a part of the useful tools for voice user interfaces.

For more details please see questions raised in the “Experimental Design” section.

Additional comments

Thanks for the opportunity to review this manuscript. The composition of the paper is commendable. In its entirety, the study and the attendant findings provide valuable commentary for the evaluation of voice user interfaces and offer insights into how SUS could be useful in the field of UX research. For the purposes of this journal, I think it can be published after addressing the comments above.

---

## Round 0.2 · accepted · Accept

The Academic Editor is not available so I have taken over handling the submission.

The original reviewers have gone through the paper again and now agree that the paper is ready for publication in its current form.

·

Basic reporting

No comment.

Experimental design

No comment.

Validity of the findings

No comment.

Additional comments

All the issues from the first review were reviewed and addressed.

·

Basic reporting

Thank the authors for taking the time to go through and answer each question. Questions regarding the graph and confidence interval have been answered. All types are fixed. The Introduction also includes a list of detailed reasons for using SUS for usability evaluation.

Experimental design

1. From lines 111 to 113: "..we recruited 16 participants (7 female and 9 male) from universities and research organizations. Their mean age was=28.43 years (SD=3.48). The inclusion criteria for participant recruitment were...". Thanks for including the participant criteria.

2. From lines 135 - 138: "To avoid preferential biases of one device over the other, we followed best practices
136 according to Dumas and Fox (2007)...". The new paragraph includes the experiment process to avoid bias.

Validity of the findings

Based on the rewritten conclusion paragraph, the author includes the benchmark and confidence interval for the findings as well as discusses the limitations of using such a scale. The findings can be considered valid as the results are obtained by conducting in-person experiments.

While the data samples are limited(16 participants), this paper can be seen as a preliminary study for adopting the SUS score for voice user interface evaluation.

Additional comments

Thanks for addressing all the comments.